# Hidden Hippos: Using Photogrammetry and Multiple Imputation to Determine the Age, Sex, and Body Condition of an Animal Often Partially Submerged

Victoria L. Inman [1],* and Keith E. A. Leggett [2]

1 Centre for Ecosystem Science, School of Biological, Earth and Environmental Sciences, University of New South Wales Sydney, Sydney, NSW 2052, Australia
2 Fowlers Gap Arid Zone Research Station, University of New South Wales, Fowlers Gap, NSW 2880, Australia
* Correspondence: victoria.inman@outlook.com

**Abstract:** Demographic Information on threatened species is important to plan conservation actions. Due to their aquatic lifestyle, the subtle nature of hippo sexual dimorphism, and their occurrence in inaccessible areas, it is difficult to visually determine hippo ages and sexes. Previously, hippo body lengths have been measured from drone images and used to estimate age. However, due to hippos' propensity to be partially submerged, it is often difficult to obtain the required measurements. We used the novel technique of multiple imputation to estimate missing body measurements. Further, we explored if male and female hippos could be differentiated in drone images based on body proportions, also examining body condition indices and how these varied seasonally. Multiple imputation increased the number of hippos that we aged threefold, and the body lengths we obtained fell within the range provided in literature, supporting their validity. We provide one of the first age structure breakdowns of a hippo population not from culled hippos. Accounting for overall size, males had wider necks and snouts than females. Hippo body condition varied seasonally, indicating responses to resources and reproduction. We provide a new technique and demonstrate the utility of drones to determine age and sex structures of hippo populations.

**Keywords:** photogrammetry; multiple imputation through chained equations; missing data; hippos; demographics; drone

## 1. Introduction

For many threatened species, there is limited information on population sizes and demographics (age structure, sex ratio, and rates of reproduction and death) [1]. Yet access to demographic data is important for conservation management, as these data can indicate population trends and resource availability [2,3], and because reproduction and death influence population sizes [4]. Examining variations in an animal's body condition can also provide valuable information, by allowing for the assessment of how that animal is responding to changes in the environment, as well as its life history (e.g., reproduction) [5,6].

The age of wild animals can rarely be accurately known and is instead often estimated through direct observation of the size of animals or the development of certain characteristics (e.g., skin colour in orangutan [7]). Direct observation allows researchers to use their experience and judgement to age animals, although this results in a subjective assessment. An alternative is to take measurements of certain body parts and compare to a known relationship with age (e.g., teeth in elephant [8]). Total body length is regularly used, given there is often a well-known, strong relationship between this variable and age [9–13]. Sexing animals is often easier; however, for species that do not exhibit sexual dimorphism, or where the dimorphism is subtle, this can also be difficult to determine in the field. There are numerous ways to assess body condition of animals, though indices should be length normalized given body measurements scale with overall body size [5,6].

Field measurements of restrained or immobilised animals can be used to take body measurements [14,15]; however, for some species, these methods are unfeasible [16,17]. Photogrammetry (measurements from images) is a viable alternative way to collect these data [18] and has been used on a range of animal species (e.g., deer [17], pinnipeds [19], elephants [20]) normally using handheld cameras. However, generally, even this method requires researchers to be close to the animal. Recently, photogrammetry using images collected from drones has been developed [5,6,15], taking advantage of the ability to capture data without being near the animal. Orthomosaics (using many images that are geometrically corrected to uniform scale) are preferred for photogrammetry as they have reduced distortion [15]; however, measurements from single images are also acceptable, especially when animals are unlikely to remain stationary [5,6,14,18].

It is not always possible to get the required body measurements from images, for example if the animal is partially obscured. There are several ways missing data can be dealt with; commonly, animals without the necessary measurements are removed from subsequent analyses [21,22] or average measurements from other animals in the study are used to replace the missing value [23]. An alternative approach, which is used in this study, is multiple imputation through chained equations. Imputation is the process of replacing missing data with values estimated based on other available data [24]. Multiple imputation through chained equations is an extension of this, where missing data are imputed numerous times until the datasets converge, which allows for uncertainty in the imputed values to be accounted for [25].

In 2006, the conservation status of the common hippopotamus (*Hippopotamus amphibius*), hereafter called hippo, was raised from 'Least Concern' to 'Vulnerable' [26], with hippos experiencing substantial declines in both numbers and geographical range [26,27]. Accurate data on the demographics of hippo populations are essential for effective conservation [26,28,29]. Due to their propensity to be at least partially submerged, and the subtle nature of hippo sexual dimorphism, it is difficult to consistently and accurately differentiate hippo age and sex categories [27,30–37]. Much of the current knowledge of hippo demographics originates from studies of hippos that were culled, and the carcasses collected and measured [38–41]. In these studies, jaw length, tooth eruption and wear patterns, eye lens size, body weight, body length, and chest girth have been used to assign hippos to 20 age classes. Actual ages (in years) were then assigned to these classes based on the estimated ages of the oldest hippos and by comparison with a limited number of animals of known age [38,40,42,43]. Most of the ageing methods used in anatomical studies (e.g., tooth eruption) are impossible to implement in field studies; however, there is potential to estimate body size in the field.

Given there is a known relationship between hippo body length and age [39,41,42,44,45], that they often occur in inaccessible areas, and their dangerous nature [46–48], hippos appear to be a suitable candidate for drone photogrammetry for ageing. Recently, Inman et al. [21] used a drone to count hippos and assign them into three age classes (adult, subadult, juvenile) based on their full length. However, even under ideal conditions, on average only 53% of hippos were able to be measured and therefore assigned into age classes. Hippos were also measured and assigned into age classes using a drone in the Democratic Republic of Congo [49], and to overcome the challenge of assessing body length of partially submerged hippos, these researchers chose to extrapolate length based on the approximate proportions of head to back, acknowledging this was subjective. They flew multiple passes over the same pod, and still often could not get measurements of all animals [49]. More recent work in South Africa used drones to classify hippos into two age classes (adult, juvenile), but detailed methodology was not provided and, therefore, we assume the authors made a subjective decision based on size [50]. Neither project attempted to sex hippos from the drone, though the latter did when conducting ground surveys, perhaps indicating these were desired data but not possible for them to collect using the drone.

Accounting for the difficulty in obtaining full body length measurements of hippos, the aim of this study was to establish a more reliable method to age hippos, even when

they were partially submerged, by using photogrammetry from drone images and multiple imputation. Further, we investigate the possibility of differentiating between adult females and males on drone images based on body measurements, and in addition assess body condition and how it varies seasonally by looking at length-to-width ratios. As far as we can determine, this paper represents the first effort to use multiple imputation to improve photogrammetry of animals, acknowledging that body length often cannot consistently be measured, and the first paper to attempt to sex and assess the body condition of hippos from a drone.

## 2. Materials and Methods

### 2.1. Study Area

This study was conducted within a 13.83 km$^2$ section of floodplain within the Abu Concession, a wildlife management area used for non-consumptive tourism, in the Okavango Delta, located within northern Botswana (Figure 1). The Okavango Delta has a semi-arid climate with rainy summers (October–April) and is subject to an annual flood event, with the flood waters originating from the high rainfall areas in Angola and moving slowly down the Okavango Delta with peak flood extent occurring in July–September [51–55]. Flooding of the Okavango Delta does not correspond with local rainfall in the rainy season; the dry season covers a high flood period initially, and then as the dry season progresses and the flood extent reduces, a period of medium-to-low flood. Therefore, we distinguish three seasons: the rainy season when the flood is low (hereafter referred to as "rainy season (low flood)"), the dry season when flooding has peaked ("dry season (high flood)"), and the dry season when flooding has begun to recede ("dry season (med-low flood)"). The onset of the rainy season varies yearly, and we categorised it as beginning with the first rains over 10 mm and ending with the last rains of the season. This study was conducted between 22 November 2017 and 19 October 2018, with the 2017 dry season (med-low flood) period occurring from October until 6 December 2017, the rainy season (low flood) occurring from 6 December 2017 until 31 March 2018, the dry season (high flood) occurring from July until September 2018, and the 2018 dry season (med-low flood) period occurring from September to 21 October 2018.

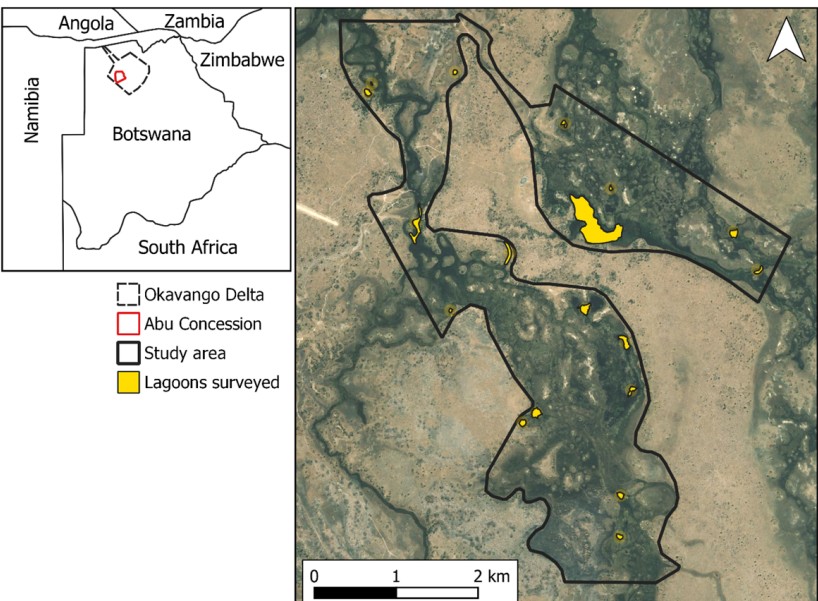

**Figure 1.** Study area and lagoons surveyed for hippopotamus (*Hippopotamus amphibius*) between 22 November 2017 and 19 October 2018 within the Abu Concession in the Okavango Delta.

### 2.2. Image Collection

Using a drone, we surveyed for hippos in all lagoons >0.001 km$^2$ in size within the study area (Figure 1), with anything less than this considered unsuitable for hippos. This was a conservative threshold to disregard lagoons, given Amoussou et al. [56] did not observe hippos in lagoons smaller than 0.2 km$^2$. In this study, the smallest lagoon occupied by a hippo (and only one solitary hippo once) was 0.0016 km$^2$, with the next smallest occupied lagoon comprising 0.0034 km$^2$, providing support for the threshold value. Suitable lagoons were identified from high-resolution drone imagery orthomosaics (refer to [57] for more information). We flew over the lagoons a total of 12 times (four times each season), each flight being two weeks ($\pm 1$ day) after the previous.

We used a multirotor DJI Phantom 4™ (4K-quality video, 12.4 MP photo, aperture of f/2.8 [58]) (DJI, Shenzhen, China), with a three-axis gimbal to stabilise the camera, and the drone controlled by a GPS-stabilised system. We recorded videos using the automatic ISO and shutter speed, which varied to promote neutrally exposed images. The camera's sensor width was 6.2 mm and focal length was 3.61 mm [59]. The drone calculates it height relative to the launch location using barometric sensors, so we launched the drone in an area with similar altitude to the survey area to ensure the correct flight height. Flights were conducted under UNSW's Animal Care and Ethics Committee (ACEC Number 17/75A); Australian Government Civil Aviation Safety Authority (Remote Pilot Licence 1023529); Civil Aviation Authority of Botswana (Remotely Piloted Aircraft Certificate Number RPA (H) 147); and The Republic of Botswana Ministry of Environment, Wildlife, and Tourism (Research Permit EWT 8/36/4 XXXIII (55)).

Outlines of the lagoons were imported into the DroneHarmony app (www.droneharmony. com (accessed on 12 November 2017)), which was used to control the drone flight during surveys. The app automatically calculated the flight routes for each lagoon to ensure the entire lagoon was captured on video with a horizontal overlap of 10% based on a height of 40 m (based on optimal flight height for surveying hippos described in [21]) and camera facing directly downward. Hippos made no obvious changes in their behaviour and did not appear to be disturbed by the drone at this height. The drone was programmed to fly (10 km/h) in transects over the lagoon, whilst continuously recording video. Video was chosen rather than still images to increase the likelihood of noticing hippos that were momentarily surfacing, as well as improving our ability to capture hippos in a suitable posture for measuring (as done for cetaceans in [6]). The optimal flight direction calculated by the app was used, except for two lagoons that were elongated and where flight direction was set for shorter transects, which allows for easier video review. The drone was operated at a minimum of 100 m from the edge of the lagoons, out of line of sight of the hippos, to avoid disturbing them. If several lagoons were in proximity, they were flown over in the same flight; otherwise, the drone was landed and then relaunched closer to the next lagoon. Surveys were conducted between 10:00 and 14:30, the optimal time determined by Inman et al. [21], except for four lagoons located near tourist lodges where flights occurred as close to optimal times as possible but varied with guest activities.

### 2.3. Photogrammetry, Multiple Imputation, and Analysis

We calculated the ground sampling distance (GSD), which represents the size of pixels, based on Equation (1):

$$\text{GSD} = \text{sensor width} \times \text{flight height} / \text{focal length} \times \text{image width} \qquad (1)$$

This was calculated for each image with height set as 40 m, and image width as 3840 or 4096 pixels (dependent on video settings used). Measuring an object of known length is a standard way of assessing the accuracy of a calculated GSD [6], and we did this using a 30 cm $\times$ 30 cm tile (see Supplementary Text S1 for details). Tile measurements obtained from drone images ranged from 29.83 cm to 30.14 cm, and we deemed this accurate enough and therefore did not correct measurements taken from the drone.

To measure the hippos from the drone images, we used the 'snapshot' function of VLC media player [60] to obtain one still image of every hippo visible in each video, by looking for instances where (from most to least important) the following occurred: most of the hippos' body was visible, the hippo was in a 'natural' resting position (stretched out with its head parallel to the water and head, neck, and body in straight alignment), and in the centre of the image. Individual images were imported into ImageJ [61], the 'set scale' function was used to input the GSD for that image and the 'straight line' function was used to take seven body measurements: back length (base of tail to neck fold), neck length (neck fold to back of ears), head length (back of ears to end of snout), body width (widest part of back), neck width (widest part of neck), forehead width (widest part between eyes and ears), and snout width (widest part of snout) (Figure 2). We only measured when the body part was clearly visible and were confident that measurements would be accurate.

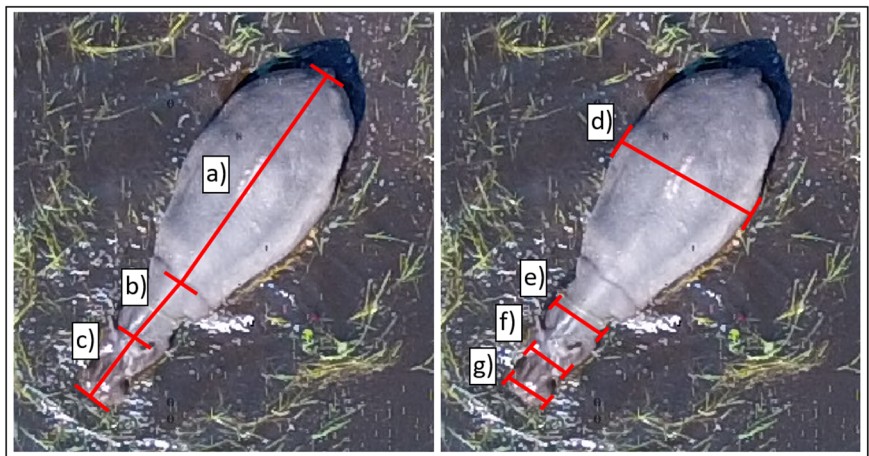

**Figure 2.** Seven body measurements were taken from drone images during surveys for hippopotamus (*Hippopotamus amphibius*) between 22 November 2017 and 19 October 2018 within the Abu Concession in the Okavango Delta: (**a**) back length (base of tail to neck fold), (**b**) neck length (neck fold to back of ears), (**c**) head length (back of ears to end of snout), (**d**) body width (widest part of back), (**e**) neck width (widest part of neck), (**f**) forehead width (widest part between eyes and ears), and (**g**) snout width (widest part of snout).

We tentatively assigned the sex of some adult hippos based on their spatial relationship with conspecifics. Hippos that were solitary (either alone in a lagoon or >100 m from another hippo) were assigned as adult males [30,41,62], and hippos with juveniles resting on them as adult females [63]. No attempt was made to sex subadult or juvenile hippos given that the sexual dimorphism at this age is more cryptic [31].

The resulting dataset had numerous missing measurements, where hippos were partially submerged, or image quality was insufficient to take either some or all measurements. Missingness patterns were investigated using the md.pattern function, and multiple imputation by chained equations was conducted to impute missing values using the mice function (MICE package [25]). We used a predictive mean matching method and set the number of multiple imputations as 70 and number of iterations as 50. We chose the number of imputations using the rule of thumb that it should be similar to the percentage of incomplete cases [64] and the number of iterations by examining trace plots to ensure variables converged. The imputations were inspected using the densityplot and plot functions. All seven body measurements were used as predictors for all other variables. Hippos with no measurements (i.e., completely submerged but still identifiable as hippos), as well as hippos that had two or fewer measurements, were excluded from imputation. However, hippos with only back length and/or body width measurements were retained, given that these were strong predictors of total length on their own.

The imputations were averaged to obtain one dataset and the total length of each hippo was calculated by adding the back, neck, and head lengths. Averaging the results of multiple imputation datasets is generally not recommended as it reduces the level of variation when performing regression [24], so while we used the averaged dataset when examining the total length for each hippo, we used all imputations for regression. We calculated the amount of variation ($R^2$) in total length that was explained by each of the seven body measurements by fitting linear regression models (lm function) separately for each measurement for both the measured and imputed data (using the pool function on all imputations). Body measurements were log transformed to achieve normality. The data almost certainly represent repeated measurements of the same individuals, though a lack of individual identification restricted our ability to account for this. Hereafter, we refer to "measured data" and "imputed data" to differentiate between the original dataset of actual measurements (with missing values) and the averaged imputed dataset (with missing values imputed).

For both measured and imputed data, hippos were assigned ages based on their total length. Martin [45] provides an equation to calculate age from body length, also providing the range of body lengths for each year of age, separately for male and female hippos (referred to as sex-dependent age/length relationship), where both sexes have the same total length for the first eight years and then diverge (with males being larger than females of equivalent age) (Supplementary Table S1). Unable to differentiate between adult males and females, we averaged the total lengths (referred to as averaged age/length relationship) (Supplementary Table S1). The difference in the size of the sexes increases with increasing age; therefore, averaging is likely to result in ageing errors only in older adults, and only adding a maximum error of four years. Based on the given ages, hippos were assigned to three age classes (juveniles, subadults, and adults), with hippos greater than four years old assigned as adults (given the lower-end estimates of age of sexual maturity [42,65–72]), hippos two to four years old were categorised as subadults and hippos less than two years old were categorised as juveniles (hippos produce a calf about once every 1.5–2 years [38,69]).

To determine if there was scope for male and female hippos to be differentiated in drone images, we examined if they had significantly different body proportions, by fitting linear regression models separately for each of the seven measurements, with sex (assigned based on spatial relationships) as an explanatory variable, as well as total length (to control for the fact that male hippos are larger overall). For this analysis, we used only the measured data (not imputed). To account for multiple testing, p values were adjusted using the p.adjust function (method = "BY"), which controls the false discovery rate [73].

To assess hippo body condition, we used the ratio of each width measurement (body width, neck width, forehead width, snout width) to total length, only using measured (not imputed) data. There were insufficient numbers of juveniles, subadults, adult females, or adult males to analyse body condition for these groups, so we only calculated body condition for non-sexed adult hippos. We examined how body condition varied seasonally, by fitting separate linear models for each width-to-length ratio, with the ratio as the response variable and season as the explanatory variable.

All statistics were conducted in R version 3.5.2 [74]. Means are reported ± standard error.

### 2.4. Validating Methods

To test the validity of using body length measurements from drone images to assign ages, and how multiple imputation of missing data might affect this, we applied the method to hippos of known age class/sex. On 14 August 2018, we visually assigned all hippos within a lagoon into three age classes (juveniles, subadults, and adults) using the method described in Inman et al. [21]. In addition, adults were classified as females using the above method (also tentatively recorded as females if a subadult was resting on them) [63]. Adult males were identified by their large size, having substantially larger necks and heads than

females, and with larger canines (visible when yawning), which results in a bulge behind their nostrils when their mouth is closed [27,31,43,75,76]. Once the visual assessment was complete, we took drone images of each hippo and calculated the same measurements as earlier, ageing each hippo using the average age/length relationship, as well as the sex-dependent relationship where possible. The age classes assigned from drone images were then compared to those we assigned visually. We then tested the accuracy of multiple imputation by randomly removing body measurements from the validation dataset using the sample function, based on the percentage that each body measurement was missing from the main dataset. This incomplete validation dataset was then joined to the measured dataset from the previous section and multiple imputation rerun using the same inputs as before. Ages and age classes were assigned based on the imputed body lengths.

## 3. Results

In total, 576 hippos were detected in drone videos (see Supplementary Table S2 for details on each flight). We were unable to take any of the seven body measurements for 11.8% of these hippos. For the remaining hippos, the number of body measurements able to be taken varied, with only 18.6% of hippos having all seven body measurements. Back length was the measurement most often missing, and forehead width was the most common measurement (Table 1). There were 141 hippos for which back, neck, and head length measurements were available, and therefore, the total length could be calculated. After removing hippos with no measurements (68) and hippos with two or fewer measurements (57), 451 hippos were included in the imputation. Density plots suggested relatively good fit of the imputed data, though imputed values for back length, body width, neck length, and neck width showed some shifting distribution patterns to lower values compared to the measured data (Supplementary Figure S1). For both datasets (measured and imputed), total length was most strongly correlated with back length followed by body width (Table 1 and Supplementary Figure S2). The order of the remaining measurements varied between the measured and imputed datasets, though neck width and length were the poorest predictors of total length in both datasets (Table 1 and Supplementary Figure S2). Hippo length ranged from 109 to 354 cm (Figure 3), which corresponds to the full hippo age range (<one year old to 45 years old) using the average age/length relationship. Of the 141 hippos for which total length could be calculated from the measured data, 5.7% were juveniles, 7.8% subadults, and 86.5% adults (Figure 3). Using the imputed data, this changed to 13.3% juveniles, 14.4% subadults, and 72.3% adults (Figure 3).

**Table 1.** Percentage of each hippopotamus (*Hippopotamus amphibius*) body measurement missing and variation in total length ($R^2$) explained by each measurement (all log transformed) for measured and imputed data collected during surveys between 22 November 2017 and 19 October 2018 within the Abu Concession in the Okavango Delta.

| Variable | % Missing | $R^2$ Measured | $R^2$ Imputed |
|---|---|---|---|
| log(Back length) | 68.7 | 0.95 | 0.95 |
| log(Neck length) | 44.1 | 0.61 | 0.72 |
| log(Head length) | 9.6 | 0.67 | 0.77 |
| log(Body width) | 65.7 | 0.83 | 0.84 |
| log(Neck width) | 43.9 | 0.63 | 0.72 |
| log(Forehead width) | 6.3 | 0.70 | 0.75 |
| log(Snout width) | 26.6 | 0.71 | 0.76 |

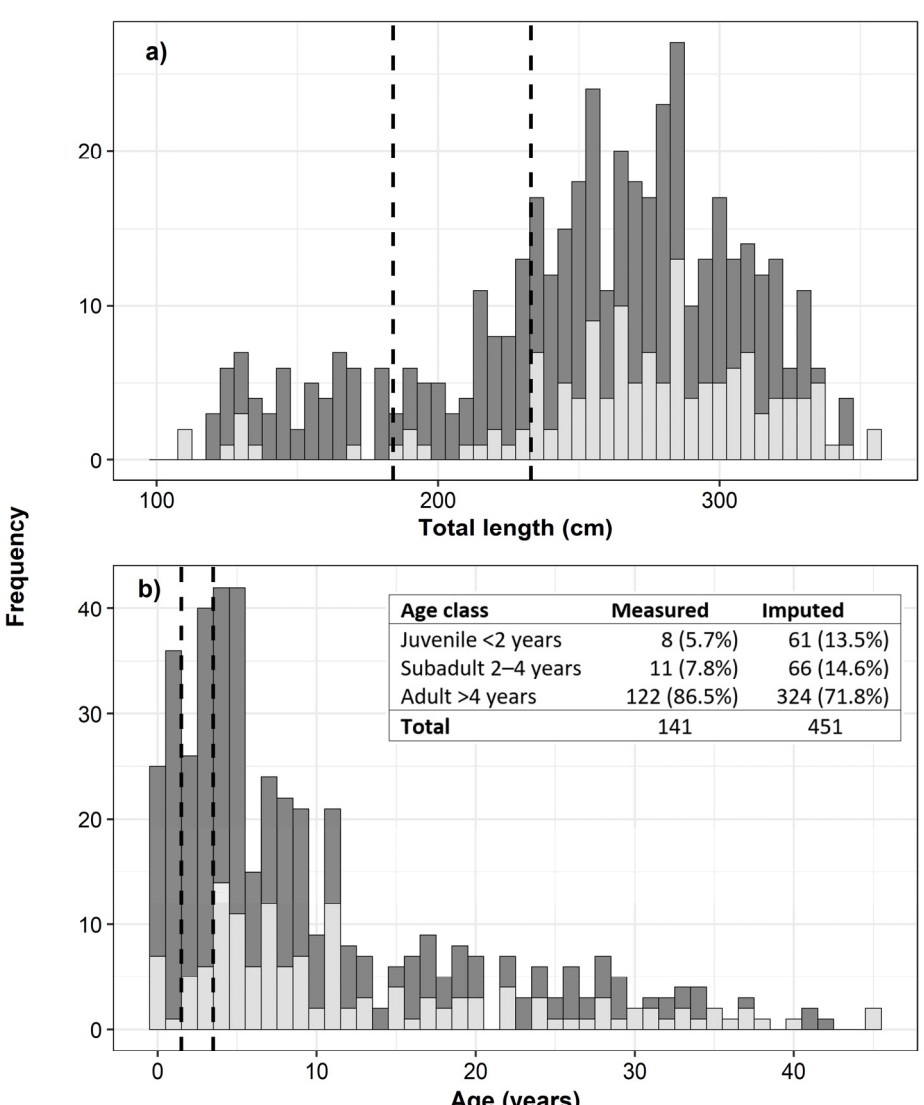

**Figure 3.** Frequency histogram of hippopotamus (*Hippopotamus amphibius*) (**a**) total length and (**b**) age for measured data (light grey) and imputed data (dark grey) and table of number and percentage of hippos in each age class, from data collected during surveys between 22 November 2017 and 19 October 2018 within the Abu Concession in the Okavango Delta. Vertical lines indicate age classes.

Of the 53 hippos that were assigned as adult males (28) and females (25) based on their spatial relationship with other hippos, 13 males and 9 females were missing at least one length measurements and therefore had to have values imputed; they were all correctly classified as adults based on the imputed values. All seven body measurements were obtained for 24 (45.3%) of the adult male and female hippos. After accounting for total length, male hippos had significantly wider necks ($t_{23}$ = 4.114, BY adjusted $p$ = 0.006) and snouts ($t_{22}$ = 3.977, BY adjusted $p$ = 0.006) than female hippos (Figure 4). There were no significant differences in back length, neck length, head length, body width, or forehead width (all $p$ > 0.05) between the sexes after controlling for total length (Figure 4). The largest adult male we measured (354.4 cm) was slightly smaller than the maximum size given for adult males (359 cm), as was the largest adult female (338.4 cm, maximum size 343 cm). Average measurements for male and female adult hippos are presented in Supplementary Table S3. Using the sex-dependent age/length relationship, adult males were on average 25 years old (range 11–42) with adult females on average 16 years old (range 4–42). Based on the measurements, we tentatively assigned hippos in the complete dataset as adult

males if they had measurements that were one standard deviation bigger than the largest female measurement or were larger than the upper total length measurement of females (343 cm [45]). This resulted in 23 hippos being assigned as male, of which 17 were 'new' males (i.e., were not assigned as males based on their spatial relationships to other hippos). None of the hippos that were assigned as male based on their size were assigned female based on their spatial relationships.

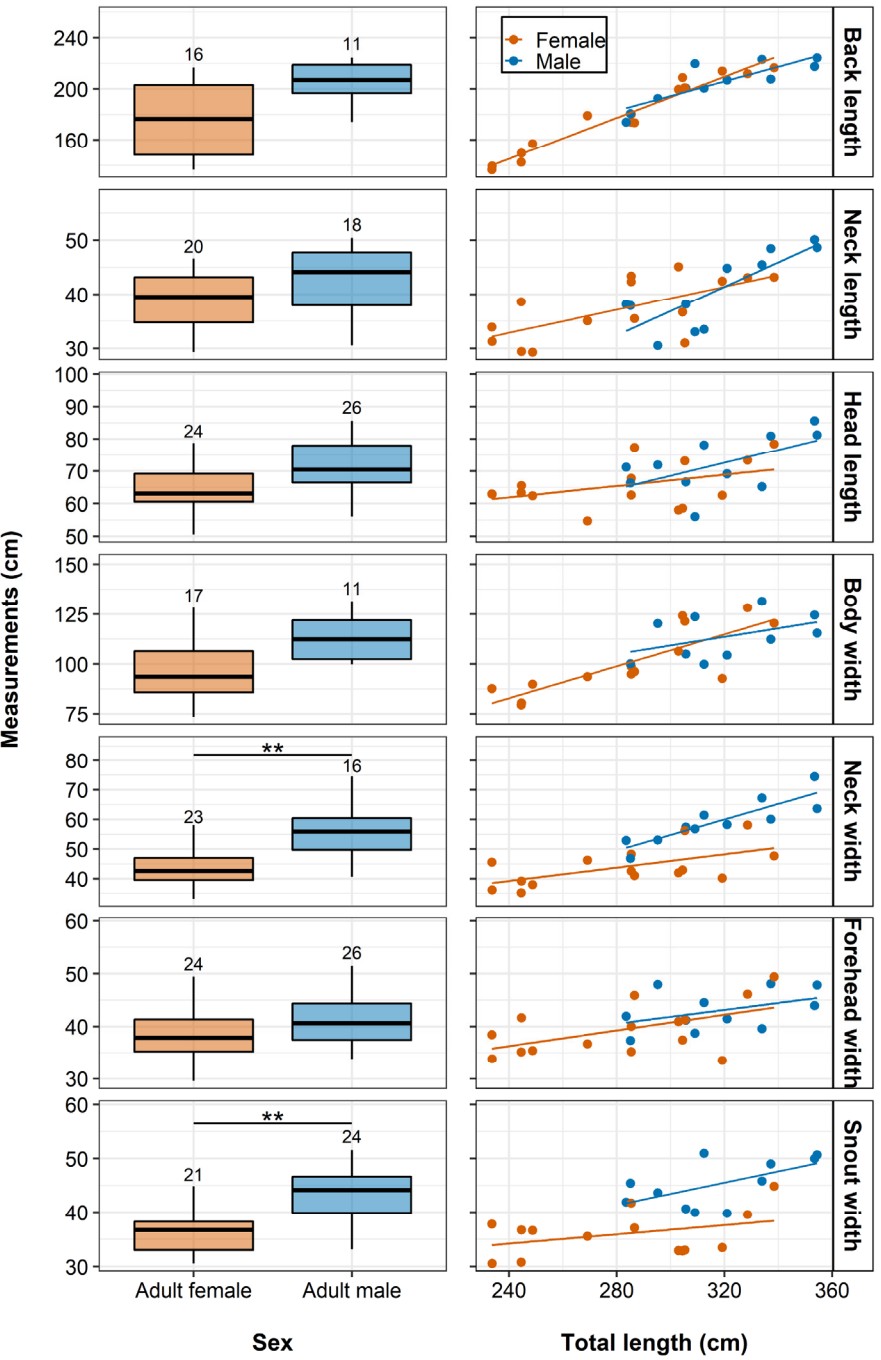

**Figure 4.** Body measurements and linear relationship between body measurements and total length of adult hippopotamus (*Hippopotamus amphibius*) based on sex. Significant post hoc pairwise comparisons identified by asterisks (**, *p*-value ≤ 0.01). Values above boxplots indicate the sample size. Data collected during surveys between 22 November 2017 and 19 October 2018 within the Abu Concession in the Okavango Delta.

The ratio of body width to total length for adult hippos varied significantly with season ($F_{2,104}$ = 8.303, *p* < 0.001), being lower (i.e., leaner) in the rainy season (low flood) than the semi-dry season (high flood) (*p* < 0.001) and dry season (med-low flood) (*p* = 0.034) (Figure 5). The other width ratios did not significantly change with season: neck width to total length ratio ($F_{2,114}$ = 1.036, *p* = 0.358), forehead width to total length ratio ($F_{2,119}$ = 1.714, *p* = 0.185), and snout width to total length ratio ($F_{2,104}$ = 1.356, *p* = 0.262).

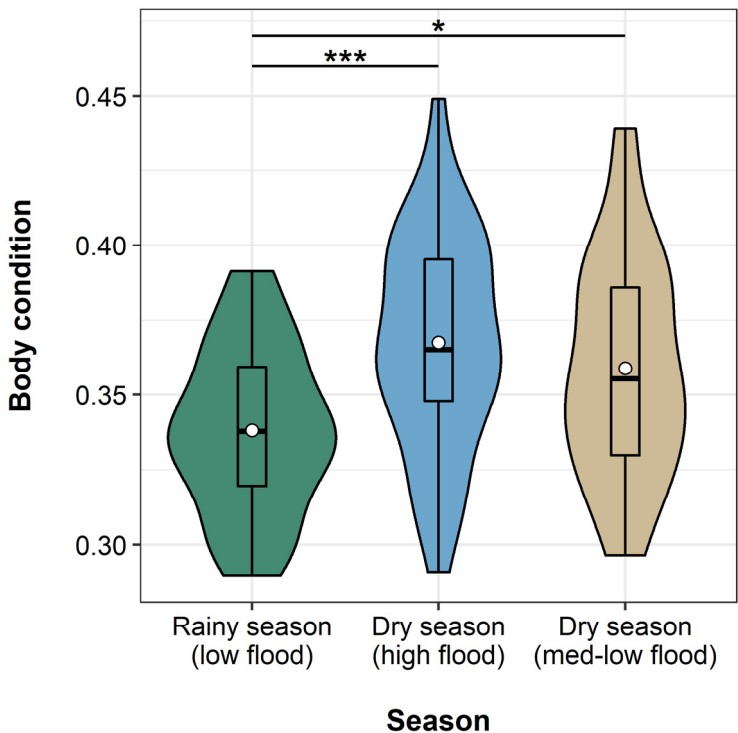

**Figure 5.** Violin and boxplots (means shown by white circles) showing seasonal changes in adult hippopotamus (*Hippopotamus amphibius*) body condition (body width:total length) using data collected during surveys between 22 November 2017 and 19 October 2018 within the Abu Concession in the Okavango Delta. Significant post hoc pairwise comparisons identified by asterisks (*, *p*-value ≤ 0.05; ***, *p*-value ≤ 0.001).

For the validation, we visually assigned age classes to all 11 hippos in the lagoon, and these matched the age classes calculated from drone measurements (Table 2). The ages calculated based on the average age/length relationship ranged from less than 1 to 38, and reassigning these based on the sex-dependent relationship (where possible) changed them by up to three years. Two adults were flagged as potential males based on the drone measurements of their neck width and length and snout width (which were more than one standard deviation bigger than the largest female measurement in this study), with one visually confirmed as an adult male at the time and the other likely to be a male, given that we confirmed that there were two adult males in this pod four days later (personal observation). After randomly removing body measurements and imputing new values, assigned ages changed by up to two years, with only one hippo incorrectly reassigned as to a different age class (subadult to juvenile).

**Table 2.** Total length, age, age classes, and sexes assigned to hippopotamus (*Hippopotamus amphibius*) using visual assessment, measured data from drone images, and from imputation. Note that imputed length, age, and age class are absent for hippos 1, 4, and 7 as randomly removed values did not affect back, neck, or head length for these hippos. Hippo 8 was the only hippo misclassified using imputed data. Data collected on 14 August 2018 within the Abu Concession in the Okavango Delta.

| Hippo | Total Length (cm) | | Age | | | Age Class | | |
|---|---|---|---|---|---|---|---|---|
| | Measured | Imputed | Measured (Average) | Measured (Sex-Dependent) | Imputed | Visual | Measured | Imputed |
| 1 | 338.87 | | 38 | 35 | | Adult male | Adult (potential male): neck width 70.3 cm neck length 54.5 cm snout width 50.0 cm | |
| 2 | 317.82 | 313.73 | 27 | - | 25 | Adult | Adult (potential male): snout width 49.2 cm | Adult |
| 3 | 296.80 | 296.15 | 16 | 17 | 16 | Adult female | Adult | Adult |
| 4 | 295.40 | | 15 | 16 | | Adult female | Adult | |
| 5 | 323.25 | 323.58 | 30 | 33 | 30 | Adult (likely female) | Adult | Adult |
| 6 | 300.06 | 299.49 | 18 | 19 | 17 | Adult (likely female) | Adult | Adult |
| 7 | 264.11 | | 7 | - | p | Adult | Adult | |
| 8 | 194.68 | 178.71 | 2 | - | 1 | Subadult | Subadult | Juvenile |
| 9 | 184.62 | 196.97 | 2 | - | 2 | Subadult | Subadult | Subadult |
| 10 | 134.26 | 135.17 | <1 | - | <1 | Juvenile | Juvenile | Juvenile |
| 11 | 121.86 | 122.90 | <1 | - | <1 | Juvenile | Juvenile | Juvenile |

## 4. Discussion

In this paper, we show that drones provide a viable and effective solution for determining hippo age through body measurements. We demonstrate the use of a novel technique (multiple imputation) for photogrammetry when not all body measurements are available, increasing the number of hippos that we were able to age from 24 to 78%. We explore the use of photogrammetry to differentiate males from females, important for an animal with subtle sexual dimorphism, and show that body condition of animals can be assessed using drones, which is valuable when determining resource availability in their environment.

The body lengths we obtained from drone images fall within the published range obtained from culled hippos [39,42], providing support to their validity. Further, measuring an object of known size from the drone provided similar values to the true size, providing a high level of confidence in subsequent measurements of hippos using this technique. There was agreement when comparing hippos of visually assigned age class/sex to those assigned from drone images. A confirmed adult male was identified as such from the drone images based on neck and snout measurements, and two confirmed females and two likely females had body measurements consistent with adult females, though these could also describe young males. Further, the random removal and subsequent imputation of values for the validation pod demonstrated the validity of imputed data; only 1 hippo out of 11 was incorrectly assigned to the wrong age class. For the remaining hippos, there was minimal difference between ages based on imputed length and those from measured length. Inaccurate ageing will mainly occur in adults, as small changes in length correspond to larger changes in age, though the age class of 'adult' is likely to remain. Previous research [49] did not attempt to validate drone-assigned age classes or confirm drone measurements against an object of known size, with other researchers [50] subjectively ageing hippos.

It was rare to obtain all seven body measurements for hippos, and the probability of obtaining a measurement varied with body part. Due to their common posture of resting with their head above water whilst submerging their body, head measurements were more frequent, although snout width was often missing. We were generally less confident taking 'width' measurements, and therefore irregularly recorded them, as there were fewer indicators that the body part was fully exposed. In contrast, all three length measurements (back, neck, and head length) had obvious indicators: the base of the tail, neck fold, ears, and nostrils. Therefore, missing length measurements are more likely related to hippo posture (e.g., a hippo with only its head and neck above water). There can be a high level of uncertainty and therefore inaccuracy associated with manual detection of the edge of an animal's body, especially when water disturbance and turbidity distort the body outline [5,6]. We attempted to minimise this by only measuring hippos when we were confident their bodies were clearly exposed. There is generally good consistency between measurements taken of the same animal by different observers (e.g., sea lions [77] or whales [78]), though it would be valuable to investigate this specifically for hippos, given the abovementioned lack of clear exposure. Back length had the highest correlation with total length, which is logical given that it is the largest measurement included in the total length calculation, but it was also the measurement most often missing. Neck measurements had the lowest correlation (despite neck length's inclusion in the calculation of total length) probably because postural changes of extending and contracting the neck can change both the width and length significantly. Lhoest [49] also suggested changes in posture and body spread may have affected body measurements. Multiple imputation allowed us to increase the number of hippos that we could assign ages to threefold. Imputing body length measurements appears to be an acceptable compromise when data are limited. However, if accuracy and completeness are priorities, then multiple flights could be conducted until all length measurements for all hippos are obtained, though this would be a resource-intensive process.

This paper represents the first attempt to differentiate male and female hippos using drone measurements. Male hippos are known for their large size [76], and this was con-

firmed in our study, with all body measurements being, on average, bigger for males than females. However, only neck width and snout width were significantly larger once the effect of total size was accounted for, a result similar to that found in culled hippos where male hippos had larger neck girth and jaw size than female hippos after accounting for total size [79]. The smallest adult female (as determined by the presence of a juvenile) we measured was four years old, and this supports estimates of age of puberty as being much earlier (3–4 years old [67,69,80]) than sometimes suggested (7–15 years old [38,40]). There were another four adult females (with juveniles) younger than seven. Assigning hippos as male or female based on their spatial relationship to other hippos is not infallible. It is unlikely, though possible, that a male hippo could have a juvenile resting on them, and therefore be incorrectly assigned as female. Likewise, females may distance themselves from their pod prior to giving birth [81], which would have them incorrectly assigned as male. Males were assigned due to being solitary, and probably represent hippos that were ejected from pods, were not strong enough to defend territories, or were trying to establish a new territory [31,76,82]; given this, we indirectly excluded dominant bulls that occurred within a pod. Peripheral males are smaller than their dominant counterparts [41], suggesting our results are conservative in terms of difference between adult males and females. Further sampling with visual confirmation of sexes could improve our understanding of the body size/sex relationship, increasing our confidence and ability to determine threshold values above which certain measurements must belong to an adult male.

Our results provide one of the first classifications of the relative age structure (actual ages rather than just age classes) of a free-living hippo population. Previous age data have generally been collected from culled hippos and there were likely inherent biases; for example, younger and smaller animals are more difficult to shoot and recover and hunters tended to focus on large pods that were likely to include more females than males, meaning these age classes/sexes are likely to be under/over-represented in samples, respectively [38,43,83]. Our results show this under-representation of young hippos in culling studies; we calculated that 13.5% and 14.6% of the population were juveniles and subadults (using imputed data), respectively, compared to 2.4–2.7% and 0.6–2.4% of culled hippos [40,83]. Examining the age classes from the measured data, most hippos were adults, with similar numbers of juveniles and subadults, but the relative percentage of both juveniles and subadults increased based on the imputed data. This suggests that younger hippos were disproportionately more likely to be missing measurements, probably due to their smaller size, which means they are often more submerged and therefore less visible, in addition their propensity to submerge quicker than adults when disturbed [50]. This pattern was also noted by Inman et al. [21], where more juveniles and subadults were able to be identified from their land surveys compared to drone surveys, because they were often not able to get full body length measurements from the drone. The percentage of hippos in each age class, for both the measured and imputed data, fall within the range of values reported elsewhere [30,36,63,84–90]. Further discussion of hippo demographic structure based on these data can be found in Inman [57].

We expected hippos would be in better condition (greater body width to total length ratio) during the rainy season, owing to more abundant and better-quality graze [91,92]; however, adult hippos were leanest in this season. This may be because the rainy season is when most female hippos give birth [38], with poorer body condition potentially due to the cost of reproduction (e.g., lactation), a pattern consistent with that seen in mature whales [5,93]. Further, hippos may have been larger in the other seasons due to the presence of large pregnant females. Seasonal changes were not apparent when examining indices for other width measurements (snout, forehead, neck), indicating that these perhaps are not fat deposit areas, and emphasizing the body width to total length ratio as a valid body condition index for hippos. Similar patterns (where seasonal changes in condition were not homogenous across the body) have been noted in whales, with no evidence of a seasonal change for measurements not associated with fat storage (e.g., the head) [5,93]. Given the variation, time of year should be considered when investigating hippo body condition,

perhaps avoiding the peak birthing period when the female body condition may be less related to resource availability. There are no existing body condition scoring systems for hippos [94], nor research detailing fat distribution in hippos, to corroborate our findings.

A detailed examination of how ageing and sexing hippos using drones compares to other methods is outside the scope of this study, although we briefly discuss the advantages and disadvantages of various methods here. Historically, the ages and sexes of hippos have been determined through direct observation [31] and examining carcasses [41]. Visual assessment or photogrammetry using camera traps is also an option, although it does not appear to have been previously used; handheld cameras are unlikely to be suitable given the dangerous nature of hippos. Importantly, both drones and camera traps allow researchers to remain at a safe distance from hippos when collecting data. Further, drones can be used in areas that are otherwise inaccessible by vehicle, unlike direct observation or camera traps. Drones also allow for surveys of multiple lagoons within a short period of time, whereas for direct observations, researchers are required to move between areas each time and using camera traps, researchers must predict where hippos are likely to be. It is easy to examine large groups of hippos using the drone as the birds-eye view allows for individuals within the middle of the pod to be identified. Although drones and associated software can be prohibitively expensive, the drone used in this study was small, relatively affordable, and commercially available, and we used open access software. However, drone flying is a skill that requires training and permits. While camera traps can also be costly, direct observation requires very little equipment. Photogrammetry from camera trap images can be used to age hippos, although it requires the additional step of determining the distance the animal is from the camera. Alternatively, both camera traps and direct observation allow for the subjective assessment of ages and sexes, which can be valuable if researchers can distinguish subtle differences. However, for many researchers, an objective, repeatable method to age and sex hippos is desired, and drone photogrammetry allows for this.

## 5. Conclusions

Drones are generally well regarded as a valuable tool for non-invasive monitoring of wildlife [95–97], including hippos [21,49,57,98–101]. This paper expands on the current use of drones generally just to count hippos, by demonstrating that it can also provide accurate estimates of hippo demographic structure and body condition, even under difficult conditions when hippos are partially submerged. For hippos, there is limited opportunity to collect these data on the ground given their dangerous nature and occurrence in difficult to access areas [102–104]. Drones provide an alternative method, with low impact on hippos [21] and improving the safety for people involved. Regular collection of demographic data would allow for temporal changes to be tracked, with these data acting as an indicator of the health of the population for this threatened species.

**Supplementary Materials:** The following supporting information can be downloaded at: https://www.mdpi.com/article/10.3390/drones6120409/s1, Text S1: Assessing the accuracy of GSD calculated for drone images; Table S1: Relationship between total length and age for male and female hippos. Averaged lengths and demographic classes added; Table S2: Number of lagoons surveyed (i.e., had surface area > 0.001 km$^2$) and occupied by hippos, and number of hippos counted in the study area for the 12 surveys; Figure S1: Density plots comparing measured data (blue) to multiple imputations (red); Figure S2: Linear relationships between body measurements (log transformed) and total length for measured data. Table S3: Average age (years) and body measurements (cm) with standard deviation for adult male and female hippos (assigned based on spatial relationships), with range in brackets. Ages calculated based on the sex-dependent relationship.

**Author Contributions:** Conceptualization, V.L.I. and K.E.A.L.; methodology, V.L.I.; software, V.L.I.; validation, V.L.I.; formal analysis, V.L.I.; investigation, V.L.I.; resources, K.E.A.L.; data curation, V.L.I.; writing—original draft preparation, V.L.I.; writing—review and editing, V.L.I. and K.E.A.L.; visualization, V.L.I.; supervision, K.E.A.L.; project administration, V.L.I. and K.E.A.L.; funding acquisition, K.E.A.L. All authors have read and agreed to the published version of the manuscript.

**Funding:** This research received no external funding.

**Data Availability Statement:** Data and code are available from the corresponding author on reasonable request.

**Acknowledgments:** Thanks to Elephants Without Borders for hosting and supporting the study, the Botswana Ministry of Environment, Wildlife, and Tourism for allowing us to conduct this research. Further, we thank Fly UAS for sponsoring Remote Pilot Licence training, Keboditse "CK" Mboroma for his assistance in the field, and Wilderness Safaris and staff at Abu and Seba Camps.

**Conflicts of Interest:** The authors declare no conflict of interest.

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
