# Peer review of "Hidden Hippos: Using Photogrammetry and Multiple Imputation to Determine the Age, Sex, and Body Condition of an Animal Often Partially Submerged"

_drones, doi:10.3390/drones6120409_

Round 1
Reviewer 2 Report
Thank you for the opportunity to review this ms, which provides clear information supporting the use of drones to assess hippo demography and body condition. Overall the ms is well written and contributes substantially to the knowledge base around using drones for ecological and conservation studies. Minor comments below.
L39. Well-known
L65. Include the Latin name of the study animal at first mention. Use the full name, then define the abbreviated one.
L94. Indicating rather than indicated
L116-120. Which months were included in each of these seasons?
L117. Using the term “rainy season” is less confusing that “wet season” when you have to consider flooding dynamics. The current seasonal terms are not clear, given that you have to include details in parentheses. Please use one term per season, e.g. Rainy, high flood, low flood.
Fig. 1. Caption should stand alone. Please include information about study species and survey period.
L125. Citation for size of lagoon suitable for hippos?
L130-136. Please include information on your drone pilot license, ethical clearance and research permit.
L141-2. Please include information as to whether hippos showed any response to drones flown at this height.
Fig. 2. Same comment as above. Include information about study species, site and period.
L195. Fewer, not less
L271. How many were considered “too few”?
Fig. 3. Same comment as for other captions.
Table 1. Same comment as for other captions.
Fig. 4. Same comment as for other captions.
Fig. 5. Looks like a violin plot rather than a box plot. What do the white circles and solid lines represent? The order in which you present seasonal data should remain consistent throughout the ms. Elsewhere you start with “wet (low flood)” so the graph should follow the same order. Same comment as for other captions.
Table 2. Same comment as for other captions.
L366. It would be more representative to include the percentage represented by that one hippo here as well.
L397. Length, not lengths
L423. Data have, not data has (data is the plural of datum)
Reviewer 3 Report
I liked the work very much. It is an important topic and an interesting. The paper is well organized and written. I have no significant comments, however, I would draw the authors' attention to the work:
Breeding colony dynamics of southern elephant seals at Patelnia Point, King George Island, Antarctica by K Fudala, RJ Bialik published in Remote Sensing 12 (18), 2964, 2020.
In which age distribution of females of southern elephant seals (Miruonga Leonina) was calculated based on the power model of female body length and the ortho obtained with use of drone. Do such formulas exist for hippos? Did the authors try to do a similar analysis? Do I encourage them to do so? Confirmation of the functionality of such equations would have tremendous application in assessing the age of entire hippopotamus populations. Despite the very substantial publication base, I would suggest considering the aforementioned work and to compare the obtained results.
